# Comparative Analysis of the Genetic Diversity of Faba Bean (*Vicia faba* L.)

Eleni Avramidou [1,2], Ioannis Ganopoulos [3], Photini Mylona [3], Eleni M. Abraham [2], Irini Nianiou-Obeidat [4], Maslin Osathanunkul [5,6] and Panagiotis Madesis [1,7,*]

1   Institute of Applied Biosciences (INAB), CERTH, 6th Km Charilaou-Thermi Road, 57001 Thessaloniki, Greece
2   Laboratory of Range Science, Faculty of Forestry and Natural Environment, Aristotle University, 55135 Thessaloniki, Greece
3   Hellenic Agricultural Organization Dimitra, Institute of Plant Breeding and Genetic Resources, 57001 Thermi, Greece
4   Laboratory of Genetics and Plant Breeding, School of Agriculture, Forestry and Natural Environment, Aristotle University of Thessaloniki, 54124 Thessaloniki, Greece
5   Department of Biology, Faculty of Science, Chiang Mai University, Chiang Mai 50200, Thailand
6   Research Center in Bioresources for Agriculture, Industry and Medicine, Chiang Mai University, Chiang Mai 50200, Thailand
7   Department of Agriculture Crop Production and Rural Environment, University of Thessaly, Fytokou St., 38446 Nea Ionia, Greece
*   Correspondence: pmadesis@certh.gr

**Abstract:** Faba bean (*Vicia faba* L.) is an important grain legume with high protein content (approximately 25–30%) and high nutritional value. It is broadly cultivated in temperate areas both for human consumption and as animal feed. According to FAOSTAT (2020), the total cultivated area of faba bean reached approximately 2.5 million ha, yielding more than 4.5 million tons. The characterization of the genetic diversity in faba bean is an important parameter for genetic and biodiversity studies, germplasm characterization, and for introducing genetic variability in plant breeding. The present study aims to assess the genetic diversity among 53 Greek, varied faba bean populations provided by the Hellenic Agricultural Organization "DEMETER" seed bank. To determine the genetic diversity of the studied populations, six SCoT DNA markers were used. A total of 114 loci were obtained with 37.95% being polymorphic and 62.05% monomorphic within or between populations. SCoT markers are a useful tool for the detection of genetic diversity among faba bean populations and encourage targeted crossing strategies. The present study is the first step towards the development of an efficient breeding program.

**Keywords:** genetic diversity; SCoT molecular markers; seed bank; population genetics; plant breeding; molecular breeding

## 1. Introduction

*Vicia faba*, commonly named faba bean, is one of the oldest crops that is cultivated worldwide, and it is native to the Mediterranean–West Asia region [1,2]. Although the exact origin of faba bean is unknown, it is believed that it was one of the earliest food legumes to be domesticated since the Neolithic period [3]. Faba bean is one of the first domesticated food legumes and has a long history of cultivation [4]. The oldest known faba bean was first identified 14,000 years ago in the southern Levant [5].

Due to the high nutritional content of its seeds, faba bean is commonly utilized as a source of protein in the Mediterranean region for both human and animal nutrition [6]. Faba beans have a protein content of approximately 29% of the dry weight [7], which is higher compared to other common food legumes and makes it one of the main sources of protein for people in the Middle East, Latin America and Africa and for livestock feed in many developed countries [8]. According to [9], faba bean includes 13–14% of the cell wall

slightly lignified, 3% saturated sugars, 6.5% oligosaccharides, and about 27–34% proteins, while starch makes up about 45% of the total dry mass. Additionally, apart from being a crucial food crop, it also contributes to the provision of animal feed and fodder and has a good impact on soil productivity for the cultivation of cereal crops [10] due to its ability to fix free nitrogen and to grow in different climatic zones [11]. In cooler climates, faba bean has advantages compared to other legumes such as soybean because it is better adapted to growth under low temperatures [12].

The use of legume crops in traditional farming systems forms a symbiosis with nodule-forming bacteria that have nitrogen-fixing ability, which provides major benefits to cropping systems and the environment and contributes to agricultural sustainability through soil improvement [4]. Compared to other grain legumes, faba bean is considered an excellent protein crop due to its ability to provide nitrogen inputs into temperate agricultural systems because of its wide adaptation [13,14].

According to FAOSTAT (2018), faba bean is the fourth most widely grown cool season legume after pea (*Pisum sativum*), chickpea (*Cicer arietinum*) and lentil (*Lens culinaris* Medik.) [15,16]. Only in 2016, 2.4 million ha of faba bean was harvested globally, and 4.5 million tons of dry grains was produced overall [17]. Faba bean production ranked seventh among all legumes crops worldwide [15].

*Vicia faba* is mainly a self-pollinating species. The authors of [18] stated that faba bean has one of the highest reported genome sizes among crop legumes, about 13,000 Mb, is a diploid with 2n = 2x = 12 chromosomes [19] and has a partial crosspollinated rate between 4 and 84% [20]. In comparison to the genome of the legume *Medicago truncatula*, this is 25 times larger [13].

Nevertheless, faba bean cultivars have antinutritional compounds such as tannins, primarily located in the seed coat, which limit their use for both human and animal consumption [21]. According to Zanollo et al. (2020), the presence of tannins decreases the digestibility and nutritional availability of protein, energy and starch in monogastric animals [21]. Faba bean tannins decrease protein digestibility in poultry due to the formation of tannin protein complexes. Tannins are also reported to reduce energy [22] and the digestibility of starch [23], but the effects were not always significant.

Analyzing genetic variation is a crucial component of genetic studies, biodiversity research, germplasm characterization and the creation of genetic variety in plant breeding. Moreover, it is important to estimate genetic variation in order to choose favorable genotypes. New techniques need to be created in order to estimate and utilize genetic variation in favor of a breeding program once it is realized that significant levels of genetic variation are not expressed in the phenotype. Recent technological developments in molecular genetics have made it possible to quantify genetic diversity at the DNA level by creating different molecular markers [2]. The major advantage of molecular markers is that they are not affected by environmental factors or by plant developmental stages [24].

Different types of markers, including isozymes, random amplification of polymorphic DNAs (RAPDs), restriction fragment length polymorphisms (RFLPs), target region amplification polymorphisms (TRAPs), sequence-specific amplification polymorphisms (SSAPs), and amplified fragment length polymorphisms (AFLPs), have already been employed to assess the genetic variability of Vicia species and *V. faba* L. populations [2]. Intersimple sequence repeat (ISSR) markers were also utilized in order to explore the genetic diversity of *V. faba* accessions [25,26].

Recently, ref. [27] outlined a quick and innovative DNA marker technology called start codon targeted (SCoT) polymorphism. This marker was developed based on the short-conserved region flanking the Adenine-Thymine-Guanine (ATG) start codon in plant genes. SCoT markers are generally reproducible, and it is suggested that primer length and annealing temperature are not the sole factors determining reproducibility [28]. SCoT markers are similar to random amplified polymorphic DNA (RAPD) and intersimple sequence repeat (ISSR) because a single primer is used as forward and reverse. Moreover, as a PCR-based gene target technique, SCoT analysis has low cost and is effective to use [28]. SCoT

primers were also used for the detection of the genetic diversity of *Vicia sativa* [29], chickpea (*Cicer arietinum*) [30], *Dendrobium nobile* L [28] and grape varieties (*Vitis vinifera* L.) [31].

The aim of the current study was to assess the genetic diversity for the first time among 53 Greek varied faba bean populations, provided by the Hellenic Agricultural Organization "DEMETER" seed bank, by using SCoT molecular markers. Detecting the genetic diversity among Greek faba bean populations is a crucial step in order to encourage targeted crossing strategies and to develop an efficient breeding program.

## 2. Materials and Methods

### 2.1. Plant Material

Twenty seeds per population, provided by the Hellenic Agricultural Organization "DEMETER" seed bank, were sown to pots in a controlled growth chamber, with a photoperiod of 14/10 h light/dark. The temperature was between 20 and 27 °C, with a mean temperature of 23 °C. Seeds were collected from different locations all over Greece (Figure 1, Table 1).

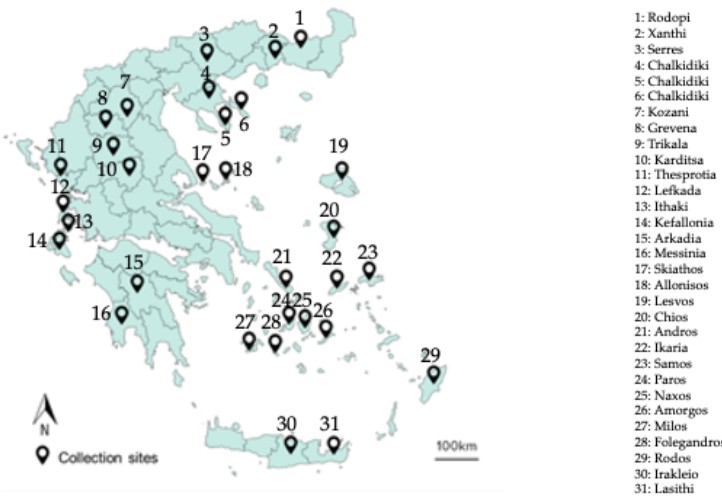

**Figure 1.** Geographical distribution of faba bean accessions from Greece.

### 2.2. DNA Isolation and Marker Analysis

Three-week-old faba bean leaves from 53 selected genotypes were collected, and one gram of fresh young leaves of 53 *V. faba* populations was ground with liquid nitrogen and saved at −20 °C (five samples per population). Total genomic DNA isolation was carried out using the modified CTAB protocol (Doyle and Doyle 1991). The protocol was modified by the technicians of the lab, and to purify DNA, only chloroform was used. Additionally, centrifuge was carried out for 10 min at $16,000\times g$. The amount of DNA was quantified by a UV-Vis spectrophotometer (Q 5000), and then the samples were diluted to a 20 ng/µL working concentration.

PCR for SCoT analysis was performed in a total volume of 20 µL containing 20 ng total genomic DNA, 100 mM of each dNTP, 1.5 mM of Mg, 10 µM of primers and 5 u/µL of Kapa taq (KappaBiosystems, Bath, UK). To study the intra- and interpopulation diversity, 6 SCoT primers (SCoT13, SCoT14, SCoT15, SCoT61, SCoT66 and ScoT33) were used for PCR amplification (Table 2). After the review of the relevant literature, specific SCoT markers were chosen because they were found to be highly polymorphic for legume species and were also used in a variety of different plant species. PCR amplifications took place in SureCycler 8800 (Agilent Technologies) as follows: initial denaturation for 5 min at 94 °C followed by 40 cycles of 30 s at 94 °C, annealing at 50 °C for 90 s, and extension at 72 °C for 90 s. A 5 min step at 72 °C was programmed as a final extension.

**Table 1.** Data of faba bean accessions provided by the Hellenic Agricultural Organization "DEMETER" seed bank.

| Sample | Code | Species | Expedition | Collection State | Collection Site |
|---|---|---|---|---|---|
| VF1 | IS-153/07 | *V. faba* | IKARIA-SAMOS | SAMOS ISL. | VOURLIOTES (37°47′6″ N, 26°50′52.8″ E) |
| VF2 | KThI-052/07 | *V. faba* | KERKIRA-THESPROTIA-IOANNINA | THESPROTIA | FROSINI (39°27′09.13″ N, 20°35′51.67″ E) |
| VF3 | IS-212/07 | *V. faba* | IKARIA-SAMOS | SAMOS ISL. | AMMOUDIES (37°37′42.08″ N, 26°47′36.79″ E) |
| VF4 | AO-060/07 | *V. faba* | AG. OROS | AG. OROS | KELI AG. MITROFANOUS (40°15.26″ N, 24°14′42″ E) |
| VF6 | KThI-051/07 | *V. faba* | KERKIRA-THESPROTIA-IOANNINA | THESPROTIA | TSAGGARI (39°25.04″ N, 20°36′39″ E) |
| VF9 | AO-033/07 | *V. faba* | AG. OROS | AG. OROS | KELI AG. SKEPIS (40°15.26″ N, 24°14′42″ E) |
| VF11 | HL-158/07 | *V. faba* | HERAKLEION-LASITHI | LASITHI | PINAKIANO (35°11′50.4″ N, 25°27′52.2″ E) |
| VF15 | HL-203/07 | *V. faba* | HERAKLEION-LASITHI | LASITHI | SXINOKAPSALA (35°3′10.8″ N, 25°52′58.8″ E) |
| VF17 | HL-077/07 | *V. sp* | HERAKLEION-LASITHI | IRAKLION | AG. VARVARA (35°9′0″ N, 25°3′0″ E) |
| VF25 | ANP-158/07 | *V. faba* | AMORGOS-NAXOS-PAROS | NAXOS ISL. | MIRISI (37°9′19.9″ N, 25°30′35.5″ E) |
| VF27 | ANP-219/07 | *V. faba* | AMORGOS-NAXOS-PAROS | PAROS ISL | LEUKES (37°03′21″ N, 25°12′31″ E) |
| VF28 | ANP-086/07 | *V. faba* | AMORGOS-NAXOS-PAROS | AMORGOS ISL. | THOLARIA (37°47′6″ N, 26°50′52.8″ E) |
| VF29 | ANP-121/07 | *V. faba* | AMORGOS-NAXOS-PAROS | NAXOS ISL. | AG. ARSENIOS (37°3′39″ N, 25°23′27″ E) |
| VF30 | ANP-066/07 | *V. faba* | AMORGOS-NAXOS-PAROS | AMORGOS ISL. | KOLOFANA (36°50′00″ N, 25°54′00″ E) |
| VF31 | ANP-169/07 | *V. faba* | AMORGOS-NAXOS-PAROS | NAXOS ISL. | APIRANTHOS (37°04′21″ N, 25°31′19″ E) |
| VF33 | ANP-171/07 | *V. faba* | AMORGOS-NAXOS-PAROS | NAXOS ISL. | APIRANTHOS (37°04′21″ N, 25°31′19″ E) |
| VF35 | ROX-031/07 | *V. faba* | RODOPI-XANTHI | XANTHI | MYRODATO (40°58′33.5″ N, 24°55′36.09″ E) |
| VF36 | ROX-145/07 | *V. faba* | RODOPI-XANTHI | RODOPI | P.KROVILI (40°57′33″ N, 25°33′30″ E) |
| VF37 | ROX-019/07 | *V. faba* | RODOPI-XANTHI | XANTHI | MELISSA (41°01′01″ N, 24°53′42″ E) |
| VF38 | ROX-003/07 | *V. sp* | RODOPI-XANTHI | XANTHI | AG. ATHANASIOS (41°02′24.00″ N, 24°46′58.08″ E) |
| VF39 | MFS-022/07 | *V. faba* | MILOS-FOLEGANDROS-SIKINOS | MILOS ISL. | ZEFIRIA (36°42′3″ N, 24°29′26″ E) |
| VF40 | MFS-060/07 | *V. faba* | MILOS-FOLEGANDROS-SIKINOS | MILOS ISL. | DASIFNOS (36°44′38″ N, 24°25′21″ E) |
| VF41 | MFS-100/07 | *V. faba* | MILOS-FOLEGANDROS-SIKINOS | FOLEGANDROS ISL. | ANO MERIA (36°38′37″ N, 24°53′00″ E) |
| VF42 | SAS-118/07 | *V. sp* | SKIATHOS-ALONISOS-SKOPELOS | SKIATHOS ISL. | KATAVOTHRA (39°10′00″ N, 23°27′00″ E) |
| VF43 | RK-104/07 | *V. sp* | RODOS-KASTELORIZO-RO | RODOS ISL | ASKLIPIO (36°4′19″ N, 27°55′46″ E) |
| VF44 | XKA-082/07 | *V. faba* | CHALKIDIKI | CHALKIDIKI | SYKIA (40°2′20″ N, 23°56′27″ E) |
| VF46 | XKA-044/07 | *V. faba* | CHALKIDIKI | CHALKIDIKI | MARATHOUSA (38°57′08″ N, 23°36′30″ E) |
| VF47 | RK-054/07 | *V. faba* | RODOS-KASTELORIZO-RO | RODOS ISL | AG. ISIDOROS (36°09′55″ N, 27°50′59″ E) |
| VF48 | XKA-093/07 | *V. faba* | CHALKIDIKI | CHALKIDIKI | AG. PARASKEVI (38°57′08″ N, 23°36′30″ E) |
| VF50 | SAS-009/07 | *V. faba* | SKIATHOS-ALONISOS-SKOPELOS | ALONNISOS ISL. | PATITIRI (39°8′41″ N, 23°51′49″ E) |
| VF51 | XKA-091/07 | *V. sp* | CHALKIDIKI | CHALKIDIKI | PALIOURI (39°56′52″ N, 23°39′53″ E) |
| VF53 | T-096/06 | *V. sp* | TRIKALA-KARDITSA | KARDITSA | NEO IKONIO (39°16′28″ N, 22°13′10″ E) |
| VF55 | T-458/06 | *V. sp* | TRIKALA-KARDITSA | TRIKALA | GAVROS (39°48′02.5″ N, 21°35′54.6″ E) |
| VF56 | P-078/06 | *V. faba* | PELOPONESE (ARKADIA-LAKONIA-ILIA-MESSINIA) | MESSINIA | KAKANA (37°18′0″ N, 21°44′27″ E) |
| VF57 | IK-095/06 | *V. faba* | ITHAKA-KEFALONIA | ITHAKI ISL. | PERACHORI (38°20′00″ N, 20°43′00″ E) |
| VF58 | P-091/06 | *V. faba* | PELOPONESE (ARKADIA-LAKONIA-ILIA-MESSINIA) | MESSINIA | VASILIKO (37°15′52″ N, 21°53′48″ E) |
| VF60 | T-336/06 | *V. faba* | TRIKALA-KARDITSA | TRIKALA | LIGARIA (39°30′38″ N, 21°42′3″ E) |
| VF61 | T-450/06 | *V. faba* | TRIKALA-KARDITSA | TRIKALA | SKEPARI (39°47′43″ N, 21°37′5″ E) |
| VF62 | IK-124/06 | *V. faba* | ITHAKA-KEFALONIA | KEFALONIA ISL. | LOURDATA (38°6′58″ N, 20°38′6″ E) |
| VF63 | X-017/06 | *V. faba* | CHIOS-LEMNOS | CHIOS ISL. | XALKEION (38°20′0.06″ N, 26°5′54.42″ E) |
| VF65 | SK-071/06 | *V. faba* | SERRES-KILKIS | SERRES | KATO POTAMIA KILKIS (40°57′24″ N, 22°56′04″ E) |
| VF66 | X-015/06 | *V. faba* | CHIOS-LEMNOS | CHIOS ISL. | XALKEION (38°20′0.06″ N, 26°5′54.42″ E) |
| VF67 | K-044/06 | *V. faba* | KOZANI-GREVENA | KOZANI | SPILIA (40°35′26″ N, 21°46′36″ E) |
| VF69 | K-034/06 | *V. faba* | KOZANI-GREVENA | KOZANI | PTOLEMAIDA (40°30′52.99″ N, 21°40′43.00″ E) |
| VF70 | M-122/06 | *V. faba* | MITILINI (LESVOS) | LESVOS ISL. | AG. PARASKEVI (39°14′53″ N, 26°16′17″ E) |
| VF71 | P-215/06 | *V. faba* | PELOPONESE (ARKADIA-LAKONIA-ILIA-MESSINIA) | ARKADIA | ARTEMISIO (37°40′34″ N, 22°22′40″ E) |
| VF72 | K-150/06 | *V. faba* | KOZANI-GREVENA | GREBENA | PONTINH (40°4′16″ N, 21° 40′36″ E) |
| VF74 | K-230/06 | *V. faba* | KOZANI-GREVENA | GREBENA | PALIOURIA (39°57′07″ N, 21°43′12″ E) |
| VF76 | IK-026/06 | *V. faba* | ITHAKA-KEFALONIA | LEFKADA ISL. | KARUA (38°45′0″ N, 20°39′0″ E) |
| VF77 | IRENA | *V. faba* | Cultivar | | |
| VF78 | DIVA | *V. faba* | Cultivar | | |
| VF79 | MELODIE | *V. faba* | Cultivar | | |
| VF80 | FAVEL | *V. faba* | Cultivar | | |

**Table 2.** Name and sequence of the SCoT molecular markers used in genetic diversity of faba bean accessions.

| Primers | Sequence 5′→3′ | Annealing Temperature | Size Range (bp) |
|---|---|---|---|
| SCoT 13 | ACGACATGGCGACCATCG | 50 | 2000–200 |
| SCoT 14 | ACGACATGGCGACCACGC | 50 | 3000–100 |
| SCoT 15 | ACGACATGGCGACCGCGA | 50 | 3000–100 |
| SCoT 61 | CAACAATGGCTACCACCG | 50 | 3000–300 |
| SCoT 66 | ACCATGGTACCAGCGAG | 50 | 4000–300 |
| ScoT 33 | CCATGGCTACCACCGCAG | 50 | 3000–200 |

Amplification products were separated by electrophoresis on 1.5% agarose gel and stained with ethidium bromide. Gel images were placed in a UVItec transilluminator, and a 100 bp or 1 Kb DNA ladder (Invitrogen, Carlsbad, CA, USA) was used as a size marker.

Binary data points denote the presence/absence of each distinguishable band across all samples for the same primer in both replicate sets of amplifications.

### 2.3. Data Analysis

SCoT markers are dominant markers, each band representing the phenotype at a single biallelic locus. Only bands that could be unambiguously scored were used in the analysis. SCoT amplified bands were scored for band presence (1) or absence (0), and a binary qualitative data matrix was formed [32].

The percentage of polymorphic loci (P), effective numbers of alleles (NE), gene diversity (expected heterozygosity, HE), Shannon's diversity index (I) and unbiased genetic distances according to [33] were calculated using GenALEx ver.6.51b2 [32]. The hierarchical distribution of genetic variation among and within populations was also characterized by analysis of molecular variance (AMOVA) [3,34] using the GenALEx ver. 6.51b2 software [32], with variation being examined among and within populations. The tests were implemented using estimates of ΦST based on distances calculated from allelic data. Principal coordinate analysis (PCoA) was used to examine the genetic structure of *V. faba* populations using GenAlEx v. 6.51b2 [32] based on the standardized covariance of genetic distance for dominant markers. Additionally, Mantel tests [35] were conducted for the comparison of genetic and geographic distances using PopTools version 3.2.5 [36] with 99 iterations.

### 3. Results

A total of 114 loci were obtained from the six selected SCoT primers. Of all loci analyzed, 37.95% were polymorphic and 62.05% were monomorphic within or between populations. Gene diversity (GD) within the studied populations of faba bean ranged from 0.222 to 0.087, with a mean of 0.149 among the populations and the Shannon index (I) from 0.127 to 0.328 with a mean of 0.220 among the populations (Table 3). Among the populations, VF37 exhibited the highest level of polymorphism (P = 57.02%) and VF78 the lowest (P = 21.05%). In addition, VF37 and VF78 showed higher and smaller levels of genetic variation for Shannon's information index (0.328 and 0.127, respectively) compared to other populations (Table 3).

The AMOVA analysis indicates that 56% of the total genetic variation was attributed to differences within populations, and the rest (44%) was attributed to differences among populations (Figure 2, Table 4).

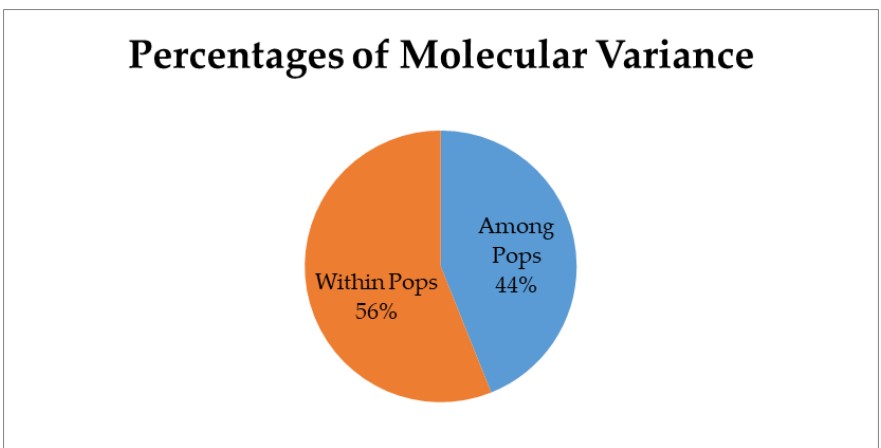

**Figure 2.** Distribution of percentages of molecular variance among fifty-three faba bean populations.

**Table 3.** Genetic diversity of fifty-three *V. faba* populations.

| Population | N [1] | NPB [2] | No. of Private Bands | P (%) [3] | Shannon Index (I) [4] | GD [5] |
|---|---|---|---|---|---|---|
| VF9 | 5 | 55 | 0 | 29.82 | 0.170 | 0.115 |
| VF11 | 5 | 64 | 0 | 34.21 | 0.197 | 0.133 |
| VF28 | 5 | 54 | 0 | 23.68 | 0.138 | 0.094 |
| VF30 | 5 | 51 | 0 | 29.82 | 0.181 | 0.125 |
| VF57 | 5 | 54 | 1 | 27.19 | 0.159 | 0.108 |
| VF72 | 5 | 51 | 0 | 22.81 | 0.135 | 0.093 |
| VF35 | 5 | 50 | 0 | 28.07 | 0.159 | 0.107 |
| VF44 | 5 | 54 | 0 | 34.21 | 0.197 | 0.133 |
| VF74 | 5 | 45 | 0 | 23.68 | 0.137 | 0.093 |
| VF77 | 5 | 53 | 0 | 28.07 | 0.166 | 0.114 |
| VF78 | 5 | 44 | 0 | 21.05 | 0.127 | 0.087 |
| VF80 | 5 | 54 | 0 | 36.84 | 0.207 | 0.139 |
| VF58 | 5 | 54 | 0 | 27.19 | 0.154 | 0.104 |
| VF62 | 5 | 58 | 0 | 39.47 | 0.235 | 0.161 |
| VF65 | 5 | 61 | 0 | 37.72 | 0.211 | 0.142 |
| VF79 | 5 | 49 | 0 | 28.95 | 0.166 | 0.112 |
| VF17 | 5 | 70 | 0 | 42.11 | 0.250 | 0.171 |
| VF31 | 5 | 68 | 0 | 45.61 | 0.260 | 0.175 |
| VF39 | 5 | 63 | 0 | 35.09 | 0.198 | 0.133 |
| VF41 | 5 | 61 | 0 | 35.09 | 0.209 | 0.143 |
| VF38 | 5 | 66 | 0 | 42.98 | 0.254 | 0.174 |
| VF27 | 5 | 59 | 0 | 38.60 | 0.216 | 0.145 |
| VF71 | 5 | 62 | 0 | 42.98 | 0.256 | 0.175 |
| VF51 | 5 | 55 | 0 | 41.23 | 0.231 | 0.154 |
| VF36 | 5 | 68 | 2 | 47.37 | 0.266 | 0.178 |
| VF46 | 5 | 67 | 0 | 38.60 | 0.225 | 0.153 |
| VF15 | 5 | 59 | 0 | 39.47 | 0.232 | 0.159 |
| VF40 | 5 | 60 | 0 | 43.86 | 0.254 | 0.173 |
| VF50 | 5 | 63 | 0 | 48.25 | 0.281 | 0.191 |
| VF1 | 5 | 60 | 0 | 36.84 | 0.212 | 0.143 |
| VF25 | 5 | 63 | 0 | 43.86 | 0.265 | 0.182 |
| VF29 | 5 | 63 | 0 | 36.84 | 0.204 | 0.136 |
| VF33 | 5 | 67 | 0 | 38.60 | 0.229 | 0.157 |
| VF42 | 5 | 66 | 0 | 43.86 | 0.259 | 0.177 |
| VF43 | 5 | 60 | 0 | 34.21 | 0.194 | 0.131 |
| VF47 | 5 | 67 | 0 | 45.61 | 0.275 | 0.189 |
| VF48 | 5 | 63 | 1 | 43.86 | 0.241 | 0.160 |
| VF55 | 5 | 57 | 0 | 34.21 | 0.206 | 0.142 |
| VF56 | 5 | 61 | 0 | 34.21 | 0.200 | 0.136 |
| VF66 | 5 | 59 | 0 | 42.98 | 0.238 | 0.159 |
| VF67 | 5 | 58 | 0 | 38.60 | 0.225 | 0.153 |
| VF76 | 5 | 62 | 0 | 46.49 | 0.278 | 0.191 |
| VF2 | 5 | 64 | 0 | 39.47 | 0.238 | 0.164 |
| VF6 | 5 | 64 | 0 | 41.23 | 0.240 | 0.163 |
| VF37 | 5 | 67 | 0 | 57.02 | 0.328 | 0.222 |
| VF70 | 5 | 76 | 0 | 48.25 | 0.273 | 0.184 |
| VF69 | 5 | 75 | 0 | 44.74 | 0.257 | 0.174 |
| VF61 | 5 | 75 | 0 | 36.84 | 0.206 | 0.138 |
| VF60 | 5 | 61 | 0 | 35.96 | 0.203 | 0.136 |
| VF63 | 5 | 72 | 0 | 46.49 | 0.278 | 0.191 |
| VF4 | 5 | 62 | 0 | 35.09 | 0.204 | 0.139 |
| VF3 | 5 | 68 | 0 | 46.49 | 0.275 | 0.188 |
| VF53 | 5 | 62 | 0 | 45.61 | 0.265 | 0.180 |
| MO | 5 | | | 37.95 | 0.220 | 0.149 [1] |

[1] N: number of individuals; [2] NPB: number of polymorphic bands; [3] P (%): percentage of polymorphic bands; [4] GD: gene diversity; I: [5] Shannon's information index.

**Table 4.** Analysis of molecular variance results for fifty-three faba bean populations based on SCoT molecular markers.

| Source | df | SS | MS | Est. Var. | % |
|---|---|---|---|---|---|
| Among Pops | 52 | 2722,906 | 52,364 | 8344 | 44% |
| Within Pops | 212 | 2256,800 | 10,645 | 10,645 | 56% |
| Total | 264 | 4979,706 | | 18,989 | 100% |
| Stat | Value | P(rand ≥ data) | | | |
| PhiPT | 0.439 | 0.010 | | | |

Genetic differentiation (Nei's genetic distance) [33] between faba bean populations ranged from 0.049 (VF3 to VF53 populations) to 0.655 (VF72 to VF39 populations) (Table 5).

**Table 5.** Pairwise genetic distances (below diagonal) between the accessions of faba bean.

| Population | VF9 | VF11 | VF28 | VF30 | - | VF63 | VF4 | VF3 | VF53 |
|---|---|---|---|---|---|---|---|---|---|
| VF9 | 0.000 | | | | | | | | |
| VF11 | 0.139 | 0.000 | | | | | | | |
| VF28 | 0.221 | 0.138 | 0.000 | | | | | | |
| VF30 | 0.201 | 0.117 | 0.161 | 0.000 | | | | | |
| - | - | - | - | - | 0.000 | | | | |
| VF63 | 0.416 | 0.392 | 0.419 | 0.370 | - | 0.000 | | | |
| VF4 | 0.567 | 0.521 | 0.607 | 0.545 | - | 0.334 | 0.000 | | |
| VF3 | 0.461 | 0.414 | 0.481 | 0.411 | - | 0.218 | 0.218 | 0.000 | |
| VF53 | 0.418 | 0.408 | 0.458 | 0.411 | - | 0.197 | 0.326 | 0.049 | 0.000 |

The UPGMA dendrogram, based on the genetic distances between populations, depicted two main clusters, which are further subdivided. Cluster I and II grouped the highest number of accessions (eight), but cluster II was divided into two subclusters. The first subcluster contained eleven accessions (VF2, VF63, VF60, VF6, VF4, VF3, VF53, VF69, VF61, VF70 and VF37). Those subclusters are also divided into two clusters. The first one encompassed sixteen accessions (VF28, VF30, VF11, VF9, VF44, VF57, VF72, VF35, VF74, VF77, VF78, VF80, VF79, VF65, VF58 and VF62). The second on encompassed thirteen accessions (VF29, VF33, VF1, VF25, VF42, VF43, VF47, VF48, VF67, VF76, VF66, VF55 and VF56) (Figure 3). However, it is not able to discriminate faba bean accessions on the basis of their geographical origin. The only obvious correlation is among the populations VF77, VF78, VF79 and VF80, which are cultivars varieties. PCoA was used to examine associations among 53 accessions. The first principal component (PC1) and the second principal component (PC2) specified the localization of individuals. PC1 and PC2 in this analysis explained 20.39% and 15.30% of the overall variability, respectively. A different perspective on the genetic distances of faba bean accessions is provided by the PCoA (Figure 4). The plant species formed clusters (Figure 3), which is in accordance with the UPGMA dendrogram. PCoA was also unable to distinguish faba bean accessions according to their geographical origin and also confirmed the results observed in the UPGMA dendrogram.

Finally, the Mantel test showed a relatively high correlation between the genetic and the geographic distances (R = 0.878, $p < 0.05$, 99 iterations).

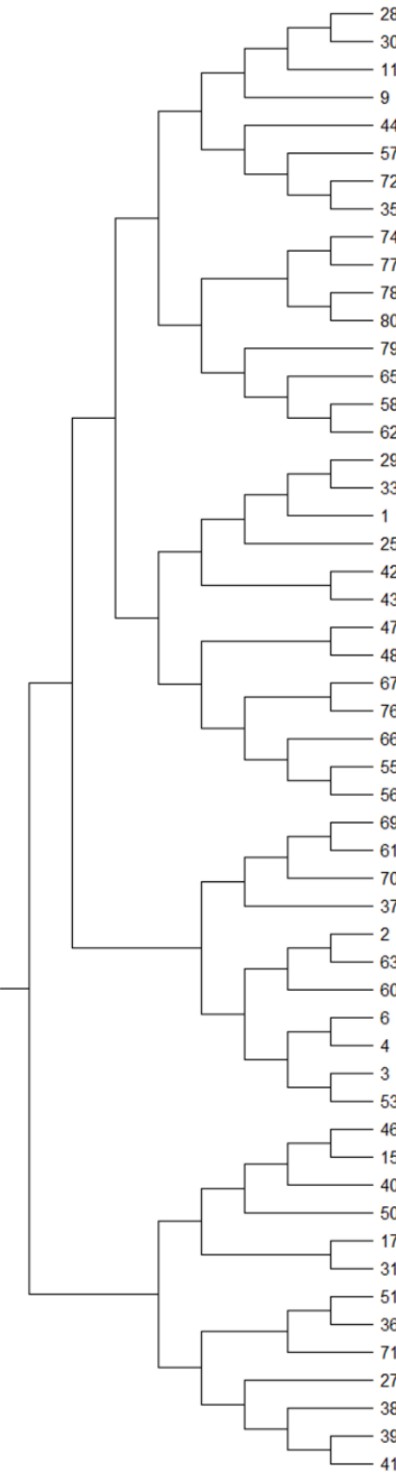

**Figure 3.** UPGMA dendrogram based on Nei's genetic distance of 53 faba bean populations.

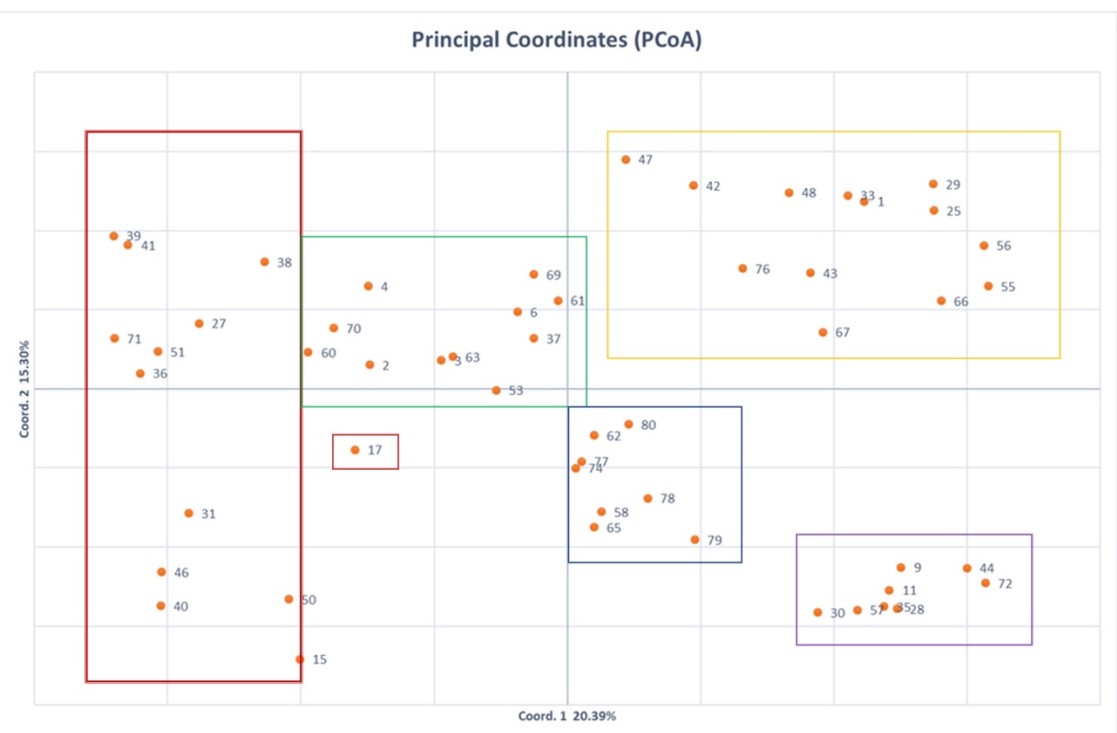

**Figure 4.** Principal coordinate analysis (PCoA) of 53 faba bean accessions using 6 SCoT markers.

## 4. Discussion

The knowledge of genetic variation within collections of Greek faba bean genetic resources is crucial for the effective conservation and utilization of these resources in breeding programs and could be dramatically enhanced by using molecular markers [2,37].

Restriction fragment length polymorphism (RFLP) markers [38], random amplified polymorphic DNA (RAPD) markers [37], amplified fragment length polymorphism (AFLP) markers [39] and intersimple sequence repeats (ISSR) markers [25] were successfully used to characterize the genetic variability among faba bean genotypes [2,40]. This is the first attempt at exploring the genetic diversity of Greek faba bean populations using SCoT molecular markers.

Several types of molecular markers were utilized in order to assess the genetic diversity of legume crops and especially faba bean. In the present study, we used six SCoT molecular markers to characterize genetic diversity in 53 Greek faba bean populations. In this study, 37.95% of the alleles were polymorphic, which is much lower compared to the results that [25] found studying the genetic diversity of 20 Greek local faba bean populations by using ISSR markers (98.9%). Moreover, ref. [26] studying the genetic diversity of Ethiopian faba bean varieties by using ISSR markers, reported that the percentage of polymorphism was 85.5%. Additionally, the percentage of polymorphism of the present study is much lower compared to SRAP markers (96% and 98.2%, respectively) [37,41], SSR markers (99% and 87.8%, respectively) [18,42] and RAPD markers (75.96%) [43]. It is obvious that different kinds of molecular markers affect the manner of polymorphism [44]. The authors of [45] stated that genetic diversity within populations is also influenced by other factors apart from the selection of molecular markers, such as long periods of low population density or gene flow rates. Additionally, low genetic differentiation within populations can result in using either pollen or seeds or long-distance gene transmission [45]. This transmission can be explained by grazing from wild or domesticated animals. Faba bean is an outcrossing crop, with the ratio of outcrossing differing between environments and genotypes. Considering that bee species are the major vectors for pollen transfer, the rate of crosspollination varies across environments, resulting in the loss of genetic diversity [46].

Furthermore, regarding the Shannon diversity index, SCoT analysis revealed an average of 0.220, while in other studies, the average was 0.63 for SSR markers [42], 0.41 for ISSR markers [26] and 0.29 for RAPD markers [43]. The results of our study are really close to the results reported by [47]. They studied the genetic diversity of some species of genus *Vicia* using ISSR and ITS molecular techniques. Shannon diversity was estimated to be 0.24. The Shannon diversity index was also estimated to be 0.262 by [48] in a study that took place in Vicia species using SSAP markers from 56 accessions from Spain and Syria.

AMOVA showed that 56% of the total genetic variation was attributed to differences within populations and that the rest (44%) was attributed to differences among populations. SCoT markers are not such polymorphisms as ISSRs, SSRs or RAPDs markers.

In contrary to the results of this study, all other studies exhibited a high level of genetic variability within populations using ISSRs (75.4%) [25] for local Greek faba bean populations, SSRs (90%) for faba bean accessions from Turkey, Sweden, Australia, Finland and Egypt [18] and RAPDs (94%) for Tunisian populations [43]. Additionally, ref. [49] reported that after evaluating the genetic diversity of 65 accessions from 12 Vicia species from the National Grass Germplasm resource bank, using SSR molecular markers, genetic variability within populations was 89% and among populations was 11%.

The same results were also found by [50], who studied the genetic diversity among 20 pea varieties and 57 accessions from wild Pisum species, and AMOVA revealed the intergroup component of variance to be 29%, while the intragroup component was 71%.

The results of the present study are not in accordance with the prementioned studies, suggesting a level of heterogeneity that is higher than expected for faba bean since it is mainly self-pollinating. Yet, crosspollination and hybrid formation has been reported to result in heterogeneous mixtures of inbreeds and hybrids [37]. In addition, the possible explanation of the observed geographical 'mixed' accessions in these populations (Figures 3 and 4) could be their moving from one place to another by human activities or by insect pollinators [51]. Grazing by wild or domesticated animals is the main human activity for the transaction of seeds and, consequently, the transaction of gene flow and exchange of genotypes in one or more cases. This is probably the reason that populations do not group according to geographical origin. This is in line with [39], who could not identify geographic partitioning of diversity in their analysis because of the low number of *V. faba* lines (*n* = 20) studied. However, the knowledge of the genetic diversity of the population under study is invaluable for the Greek Seed Bank collection as well as for faba breeders. The results obtained here provide important information on the genetic background of the Greek faba germplasm in the collection and offer at the same time the opportunity for the development of new collection and breeding strategies.

The authors of [52] studied the genetic diversity in 28 pea (*Pisum sativum* L.) genotypes using SSR markers from the Department of Genetics and Plant Breeding Institute of Agricultural Sciences in India. The 28 pea genotypes had a coefficient of genetic similarity that ranged from 0.11 to 0.73, demonstrating a high degree of genetic variety. In contrast, [53] found little variation (0.69–0.88) between cultivars of *P. sativum ssp sativum* and *P. sativum ssp arvense*. In comparison to results obtained with AFLP markers, ref. [54] showed a substantially greater similarity range (0.80–0.94) with RAPD markers in pea cultivars (0.85–0.94). Additionally, they reported that the estimated genetic diversity (0.05–0.82) among pea accessions in the study was higher than the one reported by [55] (0.0–0.66) and [56] (0.05–0.48) but comparable to that published by [57] (0.24–0.84). The greater estimated genetic distance could be attributed to variations between accessions as a result of genotype diversity in their ancestry.

According to UPGMA cluster analysis and the PCoA, the studied 53 faba bean populations were genetically evenly distributed regardless of their geographic origin, which probably suggests the mix of the genetic germplasm at some point, probably due to travel or commerce. Genetically distinguishable were the populations VF77, VF78, VF79 and VF80, which are cultivars, and they formed a group. Similar results were described in previous studies [58–60]. According to Malek et al. (2021) who studied 14 Algerian faba bean

accessions, the UPGMA dendrogram and PCoA failed to classify the accessions according to their place of origin [58]. Additionally, the authors of [60] who studied 16 Tunisia faba bean local populations also did not reveal any population and geographical area interaction. These results show that there was significant gene flow among the communities that were under study. As a result, in addition to outcrossing, human activities may affect how genetic diversity and structure are distributed in faba bean germplasm. A partial sharing of their ancestral genetic polymorphism could probably account for the overall inability of the analyzed faba bean accessions to cluster according to their place of origin. However, the Mantel test showed a relatively high correlation between the genetic and the geographic distances (R = 0.878, $p < 0.05$, 99 iterations), which is not depicted in PCoA analysis and by the UPGMA dendrogram.

Management programs of faba bean should be put in place in order to maintain its genetic diversity at the best possible level. Thus, in future programs, there should be a sampling plant set up for as many individuals as possible, with emphasis on diverse and isolated local populations. We should also stress the usefulness of and need for molecular tools and especially molecular markers as a means to monitor small populations for which pedigrees are not always available. Moreover, *Vicia faba* is a traditional legume cultivated in varied environmental conditions. Thus, it becomes even more important for *V. faba* to identify stable and high-yielding genotypes from the collection expeditions in extremely different environments, having in mind the relatively low genetic diversity of the Greek genotypes. Furthermore, it is expected that the incorporation of the genomic studies of *V. faba* along with other relative species and the discovery of SNPS, the relative high-density SNP map and the synteny among them will offer new tools of high power for the genetic studies of *V. fava.*

## 5. Conclusions

Our results with SCoT markers suggest that the 53 Greek faba bean populations utilized in this study are genetically diverse and that SCoT molecular markers are reliable and powerful tools to evaluate genetic polymorphisms and relationships among faba bean genotypes in order to develop faba bean breeding programs. However, more research should be performed with the design of an expedition for the collection of more genetic resources from more extreme environments in Greece, and furthermore, more research should study the genetic makeup using diverse germplasm, also using RILs around the world with even more markers in order to fully understand the genetic makeup of the Greek varieties and develop an efficient breeding program towards higher yield with less inputs. Yet, the results presented here give the relevant scientific community the means to already start designing and implementing an efficient breeding program.

SCoT molecular markers are innovative compared with other molecular markers since they have many advantages: they have simple operation, they do not cost much, they have an abundant rate of polymorphism, and they are more conducive to operation for molecular-assisted breeding. This study was the first attempt to explore the genetic diversity of Greek faba bean populations using SCoT markers and develop an efficient breeding program. SCoT markers revealed a relatively high percentage of heterozygosity, which facilitates the development of an efficient breeding program. Such variation has profound implications in practical breeding and should be further exploited in order to develop synthetic varieties for improving faba bean productivity.

**Author Contributions:** Conceptualization, P.M. (Panagiotis Madesis), E.A. and E.M.A.; methodology, P.M. (Panagiotis Madesis), M.O., E.A., I.G., I.N.-O. and P.M. (Photini Mylona); validation, E.A., I.N.-O., I.G., E.M.A. and P.M. (Photini Mylona); resources, P.M. (Panagiotis Madesis), P.M. (Photini Mylona) and E.A.; writing—original draft preparation, E.A. and P.M. (Panagiotis Madesis); writing—review and editing, P.M. (Panagiotis Madesis), M.O., E.A., I.G., I.N.-O. and P.M. (Photini Mylona); visualization, E.A.; supervision, P.M. (Panagiotis Madesis); project administration, P.M. (Panagiotis Madesis); funding acquisition, P.M. (Panagiotis Madesis) and E.M.A. All authors have read and agreed to the published version of the manuscript.

**Funding:** This work was partially founded by Chiang Mai University and cofinanced by the European Regional Development Fund of the European Union and Greek national funds through the Operational Program Competitiveness, Entrepreneurship and Innovation under the call RESEARCH-CREATE-INNOVATE (project code: T1EDK-04448).

**Institutional Review Board Statement:** Not applicable.

**Informed Consent Statement:** Not applicable.

**Data Availability Statement:** The datasets generated during and/or analyzed during the current study are available from the corresponding author on reasonable request.

**Acknowledgments:** We would like to thank Chiang Mai University for the support and NAGREF—Greek Seed Bank—for the seeds provided.

**Conflicts of Interest:** The authors declare no conflict of interest.

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
