# Peer review of "Comparative Analysis of the Genetic Diversity of Faba Bean (Vicia faba L.)"

_sustainability, doi:10.3390/su15021016_

Round 1

Reviewer 1 Report (Previous Reviewer 1)

Dear author

please bring new rference for FAO(2022)

Author Response

Dear author please bring new reference for FAO (2022).

Answer. We agree with the reviewer. Corrected.

Line 20. According to FAOSTAT (2020), the total cultivated area of Faba bean reached approximately 2.5million ha, yielding more than 4.5 million tons.

Reviewer 2 Report (New Reviewer)

The authors present an interesting study on analysis of the local genetic diversity of Faba bean (Vicia faba) in Greece. The bright side of the manuscript is that to provide some useful practical details on related topic. In this context, the study contributes to different fields. However, some parts of the manuscript are not easy to understand, and discussion section should be enriched. Therefore, I would like to make some suggestions to improve the quality of the paper as below:

Abstract

Lines 20-21: “Although the exact origin of Faba bean is unknown, it is believed that it was one of the earliest food legumes to be domesticated since the Neolithic period.” In my opinion, this sentence should not be in the abstract. The Authors may want to add to introduction section with related references.

Introduction

Line 37: crops -> crop

Lines 40-41: “Seeds of faba bean were identified in the southern Levant and they are aged 14,000 years ago” -> “The oldest known faba bean was first identified 14000 years ago in the spouthern Levant.” I think, this sentence would be better fit here (or similar sentence).

Line 77: According to [20]-> According to Zanollo et al (2020) the presence of tannins decreases the digestibility and nutritional availability of protein, energy and starch in monogastric animals [20].

Materials and Methods

Line 136: 2.2. DNA isolation and markers analysis –>

Line 139: -20ο C-> -20οC

Line 140: “Total genomic DNA isolation was carried out using the modified CTAB protocol”. Did you modify the protocol? If yes, please clarify how.

Lines 145- 150: Please rephase this paragraph since it is not easy to understand.

Did you develop your own primers? or ?

Lines 166: A reference would be better here.

Discussion

The Discussion section should be enriched with a more theoretical interpretation and relating the present results with additional concepts. For instance, the study results can be discussed in the framework of local genetic diversity of closely related plant species in broader context.

Lines 316-317: “Fifty- three populations were evaluated by using SCoT molecular markers in order to study their genetic diversity” I think, this sentence is unnecessary in here since mentioned in different part of the manuscript.

Lines 278-284: Please rephase this paragraph with separate short sentences since it is not easy to understand.

Line 299: According to [48] -> According to Malek et al. (2021) …….[48].

Lines 311-314: “Our results with SCoT markers suggest that the 53 Greek faba bean populations utilized in this study are genetically diverse and SCoT molecular markers are reliable and powerful tool to evaluate genetic polymorphisms and relationships among faba bean genotypes in order to contact faba bean breeding programs.”. I think this sentence should be in the conclusion section.

Conclusion

Limitations of the study should be given.

Line 317-348: Please re-check here.

Author Response

Reviewer 2

Comments and Suggestions for Authors.

The authors present an interesting study on analysis of the local genetic diversity of Faba bean (Vicia faba) in Greece. The bright side of the manuscript is that to provide some useful practical details on related topic. In this context, the study contributes to different fields. However, some parts of the manuscript are not easy to understand, and discussion section should be enriched. Therefore, I would like to make some suggestions to improve the quality of the paper as below:

Abstract

Lines 20-21: “Although the exact origin of Faba bean is unknown, it is believed that it was one of the earliest food legumes to be domesticated since the Neolithic period.” In my opinion, this sentence should not be in the abstract. The Authors may want to add to introduction section with related references.

Answer. We agree with the reviewer. Corrected.

Line 38. Although the exact origin of Faba bean is unknown, it is believed that it was one of the earliest food legumes to be domesticated since the Neolithic period (Torres et al. 2006).

Introduction

Line 37: crops -> crop

Answer. We agree with the reviewer. Corrected.

Lines 40-41: “Seeds of faba bean were identified in the southern Levant and they are aged 14,000 years ago” -> “The oldest known faba bean was first identified 14000 years ago in the southern Levant.” I think, this sentence would be better fit here (or similar sentence).

Answer. We agree with the reviewer. Corrected.

Line 77: According to [20]-> According to Zanollo et al (2020) the presence of tannins decreases the digestibility and nutritional availability of protein, energy and starch in monogastric animals [20].

Answer. We agree with the reviewer. Corrected.

Materials and Methods

Line 136: 2.2. DNA isolation and markers analysis –> 

Line 139: -20ο C-> -20οC

Answer. We agree with the reviewer. Corrected.

Line 140: “Total genomic DNA isolation was carried out using the modified CTAB protocol”. Did you modify the protocol? If yes, please clarify how. 

Answer. The protocol was slightly modified by the technicians of the lab and to purify DNA only chloroform was used. Additionally, centrifuge was carried out for 10 min at 16000 x g.

Lines 145- 150: Please rephase this paragraph since it is not easy to understand. 

Did you develop your own primers? or? 

Answer. To study intra- and interpopulation diversity 6 SCoT primers (SCoT13, SCoT14, SCoT15, SCoT61, SCoT66 and ScoT33) were used for PCR amplification (Table 2). After the review of the relevant literature, specific SCoT markers were chosen because they were found to be highly polymorphic for legumes species, and were also used in a variety of different plant species.

We did not develop our new markers. We studied the relevant literature (genetic diversity, legumes, SCoT molecular markers) and we did a lot of experiments (PCR) in our lab in order to chose the specific SCoT markers.

Lines 166: A reference would be better here. 

Answer. We agree with the reviewer. Corrected.

Discussion

The Discussion section should be enriched with a more theoretical interpretation and relating the present results with additional concepts. For instance, the study results can be discussed in the framework of local genetic diversity of closely related plant species in broader context.

Answer. We enriched the discussion with literature which concerns genetic diversity from other legumes species such as Pisum sativum or Vicia sativa. However, taking into account that the title of this research is “Comparative analysis of the genetic diversity of faba bean (Vicia faba L.)” we preferred to focus on this specific specie. Furthermore, we added new text in the discussion section to a more theoretical interpretation of the results as suggested.

Lines 316-317: “Fifty- three populations were evaluated by using SCoT molecular markers in order to study their genetic diversity” I think, this sentence is unnecessary in here since mentioned in different part of the manuscript.

Answer. We agree with the reviewer. Corrected.

Lines 278-284: Please rephase this paragraph with separate short sentences since it is not easy to understand.

Answer. AMOVA showed that 56% of the total genetic variation was attributed to differences within populations and the rest (44%) was attributed to differences among populations. SCoT markers are not such polymorphism as ISSRs, SSRs or RAPDs markers.

So, in contrary with our results, all the other studies exhibited a high level of genetic variability within populations, using ISSRs (75.4%) [24] for local Greek faba bean pop-ulations, SSRs (90%) for faba bean accessions from Turkey, Sweden, Australia, Finland and Egypt [17] and RAPDs (94%) for Tunisian populations [43].

Line 299: According to [48] -> According to Malek et al. (2021) …….[48]. 

Answer. We agree with the reviewer. Corrected.

Lines 311-314: “Our results with SCoT markers suggest that the 53 Greek faba bean populations utilized in this study are genetically diverse and SCoT molecular markers are reliable and powerful tool to evaluate genetic polymorphisms and relationships among faba bean genotypes in order to contact faba bean breeding programs.”. I think this sentence should be in the conclusion section.

Answer. We agree with the reviewer. Corrected.

Conclusion

Limitations of the study should be given.

 Answer. Limitations are  now included in the conclusion section.

Line 317-348: Please re-check here.

Answer. We agree with the reviewer. Corrected

This manuscript is a resubmission of an earlier submission. The following is a list of the peer review reports and author responses from that submission.

Round 1

Reviewer 1 Report

I have checked the manuscript. It is well written

Author Response

Thank you very much for the time that you spend in order to read and evaluate our study.

Reviewer 2 Report

The manuscript presents a genetic study of faba bean leaves from 53 genotypes. The topic is relevant to the Sustainability journal, and it could be interesting to their readers. However, I have some questions to their methodology, and their conclusions are not tightly linked to their observations. As such, I'd recommend the authors review their data again and resubmit it after some substantial revisions. 

1. Figures 4 and 3 are not explained clearly in the text. They simply claim that "the plant species formed 197 clusters (Figure 3) which is in accordance with the UPGMA dendrogram." I don't really see it from the figures. Can authors compare the classification labels and see how they match to each other?

2. Figure 2 is completely redundant to Table 4. What does the molecular variance mean here? And why does it matter? The similar questions would also apply to the description of Table 5. Why and what do these distances mean? Are you getting the same conclusions as you would get from PCoA and dendrogram plots which were based on distances too? 

3. Without examinations of the analyses, the authors jumped into several conclusions - a. the populations were genetically evenly distributed regardless of their geographic origin; b. they are genetically diverse and c. SCoT markers are reliable and powerful tool to evaluate polymorphisms ... Again, none of these can be inferred from the data analyses presented in this paper. Maybe the authors should attempt to integrate the Table 1 and Figure 3/4 or look into the correlations between the distance matrices (geologically and genetically). 

The authors tried to support their conclusions with literature support, which is a good pathway, but a better explanation will be necessary especially regarding the inconsistencies in the discussions at lines 226, 238, and 246. 

Author Response

In advance, we would like to thank you for your time to read and evaluate our work.

We claimed that plant species formed 8 clusters, not 197. And we can compare the classification labels and see how they match to each other. For example, in the UPGMA dendrogram you can see that populations VF28, VF30, VF11, VF9, VF44, VF57, VF72 and VF35 formed one cluster. If you also see Figure 4, (PCoA analysis) these populations are grouping together

Molecular variance explains the differentiation among and within populations. We have five samples per population and it is really interesting to investigate genetic variability not only between 53 populations, but also between those 5 individuals per population.

Nei’s genetic distance is a powerful tool for measuring genetic differentiation between species or between populations within a species. If we take into account that in this study, we have 53 populations of Faba bean from different sites of Greece is very interesting to examine whether these populations are related to each other, are they genetically close or not.

The conclusions we get are really different. In the case of Nei’s genetic distance we examine the genetic distance and differentiation between populations. UPGMA dendrogram and PCoA analysis reveal groups and how close or far genetically these populations are.

a and b. Yes, the populations were genetically evenly distributed regardless of their geographic origin and their also genetically diverse.

c. According to our results, not only from this study, SCoT markers are reliable markers to revile the genetic polymorphisms and they can show and identify genetic differences. And we strongly believed that you can infer all these from the AMOVA analysis and from Table 3. If we attempt to integrate Table 1 with the Figures 3 or 4, the readers may be confused.

Line 226. [43] referred that genetic diversity within populations is also influenced by other factors except of the selection of molecular markers, such as long periods of low population density or gene flow rates. Probably, if we take into consideration that the populations were genetically evenly distributed regardless of their geographic origin and their also genetically diverse, there is a gene flow as we can interpret from UPGMA dendrogram and PCoA analysis. In which way these transaction takes place its unknown, but it can be attributed to human activities like grazing or pollen according to literature.

Line 238. AMOVA showed that 56% of the total genetic variation was attributed to differences within populations and the rest (44%) was attributed to differences among populations. SCoT markers are not such polymorphic as ISSRs or SSRs and we also have to take into consideration that we have a very big sample of population from different sites of one country.

Line 246. Also, the possible explanation of the observed geographical ‘mixed’ accessions in these populations (Figures 3 and 4) is that these accessions moved from one place to another with human activities or by insect pollinators.

Grazing by wild or domesticated animals is the main reason for the transaction of seeds and consequently transaction of gene flow.

Reviewer 3 Report

The researchers screened 53 Taba bean genotypes in Greece with 6 SCoT DNA markers. I found the main research idea really interesting.  As we know from figure 1, the sampling area is very wide and variations of genotype are enough for evaluation of genotype x environment relation among the results. Especially using the figure 3 dendrogram. The first point, authors need to tag sample codes on the Figure 1 map so that we can know the location of the samples. In the second point, authors need to find and discuss genetic relations among for example Rhodes, Samos, Lesvos, Kefallonia islands, and the mainland. Without these points, the research will be half. I suggest the researcher re-evaluate the results at this point.

Author Response

Thank you very much for the time that you spend in order to read and evaluate our study. 

The location of samples is shown in Table 1. If we tag them also on Figure 1 it will be very confusing for the reader

Geographical isolation is not reflected in the results of this study mainly due to the transaction of the seeds from the mainland to the islands and vice versa. This transaction allows mixed accessions through gene flows. But we did not distinguish populations which are geographical isolated, and something like that it is not also obligated.

Round 2

Reviewer 2 Report

This manuscript could potentially be very interesting given the amount of genetic data included in the work, which was the reason why I think this is worth reconsideration after major revisions. However, in the latest version, the authors didn't improve any of their analyses, with only very basic distance-based and variance analyses. As I opined in the first report, these analyses do NOT support their conclusions. And without any significant changes to their current analyses and any deep analyses incorporating with the geographic distribution, I don't think this manuscript would be of any interest to the readers.

A long list of questions still remains. Why are there 8 clusters? Are they associated with geographical locations? Why are there 7 seeds from a different species, V.sp? Is your PCoA (or sometimes PCA) analysis principal component analysis or principal coordinate analysis? Are there 8 clusters in the PCoA plot? Why are your Shannon index and AMOA variance different from the other studies? Why would the heterogeneity of your data be higher than the others? Why do VF77 to 80 form distinguishable group while the others not?  

Reviewer 3 Report

The authors did not respond to my comments. The samples need to be examined regarding sampling site vs. genetic similarity.  This version of the study lacks information. I advise against accepting this version.